# Vitamin D Deficiency During Pregnancy Is Associated with Postpartum Depression: A Cohort Study in Southern Brazil

**DOI:** 10.3390/nu17233649

**Published:** 2025-11-21

**Authors:** Luis Otávio Lobo Centeno, Aline Longoni, Jéssica Puchalski Trettim, Isabela Thurow Lemes, Andressa Schneider Lobato, Nathália Passos Moura, Djiovana Zanini, Thiago Falson Santana, Eduarda Neutzling Drawanz, Fernanda Teixeira Coelho, Mariana Bonati de Matos, Luciana de Avila Quevedo, Gabriele Ghisleni, Diogo Onofre Souza, Ricardo Tavares Pinheiro, Adriano Martimbianco de Assis

**Affiliations:** 1Graduate Program in Health and Behavior, Center of Health Sciences, Laboratory of Clinical Neurosciences, Catholic University of Pelotas (UCPel), Pelotas 96015-560, RS, Brazil; 2Graduate Program in Biological Sciences: Biochemistry, Institute of Basic Health Sciences, Federal University of Rio Grande do Sul (UFRGS), Porto Alegre 90035-003, RS, Brazil

**Keywords:** vitamin D, postpartum depression, pregnancy, risk factors, mental health, vitamin deficiency

## Abstract

Background/Objectives: Postpartum depression (PPD) represents a major public health issue, with a direct impact on the quality of life of the mother–infant dyad. 25-hydroxyvitamin D (25(OH)D), hereafter referred to as VitD, has been suggested to exert protective effects on mood regulation. However, current findings remain inconsistent. This study aimed to assess the association between gestational VitD deficiency (≤19.9 µg/mL) and the diagnosis of PPD three months after delivery. Methods: This longitudinal study followed mother–child dyads in the city of Pelotas, RS, Brazil. A total of 983 pregnant women were initially recruited, of whom 713 had complete data available for this analysis. Blood samples were collected up to 24 weeks of gestation for subsequent measurement of serum VitD levels using chemiluminescence, and PPD diagnosis was established using the Mini International Neuropsychiatric Interview (M.I.N.I. Plus). Logistic regression models were applied and adjusted for potential confounders, such as maternal age, socioeconomic status, and history of depression during pregnancy. Results: In the adjusted model, deficient serum VitD levels were associated with a two-fold-higher likelihood of PPD diagnosis compared to insufficient/sufficient VitD levels (≥20 µg/mL) (OR = 2.0; 95% CI 1.0–4.2; *p* = 0.049). Conclusions: These findings highlight the potential role of VitD in maternal mental health and support the importance of monitoring VitD status during pregnancy. From a public health standpoint, ensuring adequate vitamin D levels in prenatal care may contribute to reducing the burden of postpartum depression.

## 1. Introduction

Postpartum depression (PPD) is a severe mental health condition characterized by clinical symptoms such as depressed mood; loss of interest; sleep and appetite disturbances; fatigue; feelings of worthlessness or guilt; difficulty concentrating; irritability; anxiety; and, in severe cases, suicidal ideation. This condition may persist for up to one year after childbirth [1,2]. Its prevalence ranges from 5% to 20% worldwide and from 12% to 25% in Brazil [3]. The main risk factors associated with PPD are low socioeconomic status, unplanned pregnancy, marital conflicts, lack of social support, maternal age at the extremes, low self-esteem, history of depression, and recent psychological stressors [4,5,6].

However, the pathophysiological mechanisms underlying PPD are multifactorial and remain poorly understood [7,8,9]. In this context, 25-hydroxyvitamin D (25(OH)D), here referred to as VitD, has emerged as a potential modulating factor of mental health during the perinatal period [10]. Evidence suggests that VitD exerts antioxidant, anti-inflammatory, and neuroprotective effects. Moreover, it influences the HPA axis by regulating cortisol and modulating serotonin levels. These mechanisms indicate that this molecule may play a protective role in PPD development [11,12].

Despite the consistent theoretical background, empirical evidence remains limited. Several studies suggest an association between VitD deficiency and PPD; however, they present relevant methodological weaknesses. Few studies have sufficiently large sample sizes to evaluate VitD levels during pregnancy and follow women longitudinally in the postpartum period. Moreover, most studies rely solely on screening instruments for depressive symptoms rather than on more accurate clinical diagnostic methods for identifying PPD. In this study, we employed a structured clinical interview for PPD diagnosis, representing an important methodological strength. Recent systematic reviews have reported weak correlations between VitD levels and depressive symptoms, highlighting significant gaps in current knowledge. Therefore, further investigations with stronger methodological designs are essential to clarify whether gestational VitD deficiency is indeed associated with the occurrence of postpartum depression [13,14].

Therefore, this study aimed to investigate the association between maternal vitamin D deficiency during pregnancy and the occurrence of clinically diagnosed postpartum depression three months after delivery, using a longitudinal design with rigorous control for confounding factors.

## 2. Materials and Methods

### 2.1. Ethical Considerations

This study was approved by the Research Ethics Committee of the Catholic University of Pelotas (Certificate of Presentation for Ethical Consideration—CAAE 47807915.4.0000.5339, report no. 1174221). All participants signed an informed consent form, agreeing to take part in this study.

### 2.2. Design

This is a longitudinal sub-study nested within a cohort study titled “Maternal Neuropsychiatric Disorders in the Pregnancy-Puerperal Cycle: Early Detection and Intervention and Their Consequences for the Family Triad” [15]. The initial recruitment of pregnant women took place between 2016 and 2018 in all households across 244 of the 488 census tracts that comprised the urban area of the city, according to the 2010 Census Framework of the Brazilian Institute of Geography and Statistics (IBGE). All women up to 24 weeks of gestation who resided in a randomly selected tract were considered eligible. The pregnant women were assessed at their homes at the time of recruitment, totaling 983 participants in the first stage of this study. Further details on the sampling process can be found in previous publications [15,16]. Between 2017 and 2019, at three months postpartum, all mothers who had participated in the initial assessment phase were contacted and invited to participate in the second phase. Women with visual impairments, as well as those unable to understand and/or respond to the questionnaire, were excluded from the sample.

Of the 983 women evaluated at baseline (pregnancy), 755 participated in the postpartum evaluation, representing a 23.2% loss to follow-up. Of these 755, 713 women had complete information on the variables of interest (vitamin D levels during pregnancy and postpartum depression investigation), corresponding to 72.5% of the initial sample of pregnant women recruited.

### 2.3. Instruments

The presence of postpartum depression (PPD), assessed in the first stage (during pregnancy) and in the second stage (postpartum), was evaluated using module “A—Depressive Episodes” of the Mini International Neuropsychiatric Interview (M.I.N.I. Plus), a brief structured diagnostic interview developed to investigate major psychiatric disorders according to the criteria of the Diagnostic and Statistical Manual of Mental Disorders (DSM-IV) [17]. In this study, the Brazilian version of the M.I.N.I. Plus was used, which has demonstrated satisfactory reproducibility and semantic adequacy for the national context [18]. The interviews were conducted by undergraduate students in the health field, previously trained by experienced professionals, ensuring standardization in administration and minimizing measurement bias.

The socioeconomic status of the participants was assessed using the Brazilian Economic Classification Criteria of the Brazilian Association of Research Companies (ABEP), an instrument widely used in population-based and health research in the country [15]. Socioeconomic classes were grouped into three analytical categories to facilitate statistical analysis and the epidemiological interpretation of the findings: A + B (high levels), C (intermediate levels), and D + E (low levels).

The nutritional status of the pregnant women was assessed according to Atalah et al. [19], accounting for gestational age and each pregnant woman’s current body mass index (BMI). This information was used to determine the ideal weight gain for the period. The classification is based on the BMI X gestational age curve, classifying nutritional status into low weight, normal weight, overweight, or obesity. For our study, the continuous BMI score was used [20]. The BMI was calculated as weight/height^2^ (kg/m^2^), with weight measured with an anthropometric scale and height measured with a stadiometer.

Maternal variables included age (subsequently grouped into tertiles), self-reported ethnic group (White, Black, and Other), self-reported use of vitamins since the beginning of pregnancy (yes/no), self-reported weekly psychological or psychiatric treatment during pregnancy, and season of the year.

### 2.4. Collection and Processing of Blood Samples

During the first stage of this study (pregnancy), 10 mL of blood was collected by venipuncture and centrifuged at 3000× *g* for 10 min. The resulting supernatant was transferred to tubes and stored at −80 °C until VitD analysis in a certified clinical laboratory.

### 2.5. Measurement of Serum VitD (25(OH)D)

Serum 25(OH)D concentrations (ng/mL) were measured using the microparticle chemiluminescence method according to the manufacturer’s instructions using Atellica^®^ IM (Siemens, Erlangen, Germany). Atellica^®^ IM is an accurate and precise assay with a functional sensitivity of ≤3.0 ng/mL and inter-assay imprecision of ≤20% [21]. This methodology has been certified since 2018 for total 25(OH)D testing by the Vitamin D Standardization Certification Program of the Centers for Disease Control and Prevention (CDC) [22]. Serum 25(OH)D concentrations were classified using cut-off values into deficient (<19.9 ng/mL) and not deficient (≥20.0 ng/mL) [23]. Vitamin D deficiency was defined as serum 25(OH)D levels < 20 ng/mL, according to widely accepted clinical practice guidelines, which establish this value as the threshold for maintaining bone and metabolic health [23,24].

### 2.6. Data Processing and Statistical Analysis

Potential confounders for the multivariable logistic regression model were selected based on the theoretical framework consulted and the scientific and clinical expertise of the research team. The final adjusted model included the following confounders: maternal age, socioeconomic status, and history of depression during pregnancy. The model fit was assessed using the Hosmer–Lemeshow test, which indicated good adequacy across all six stages of the automatic variable selection procedure (*p*-values ranging from 0.741 to 0.863).

Regarding missing data, the analytical sample included only women with complete information for the main exposure (serum vitamin D levels) and the outcome (postpartum depression); therefore, these primary variables had no missing values. For the remaining covariates, missing values were identified and coded in the statistical software (IBM SPSS 26.0), and the analyses presented already account for missing data.

Data were collected through structured questionnaires, subsequently coded, and double-entered into EpiData 3.1 to check for inconsistencies. Statistical analyses were performed using IBM SPSS 26.0. Sample characteristics were described using absolute and relative frequencies for categorical variables. Bivariate associations were examined using the chi-square test, while multivariable analyses were conducted using logistic regression (Backward Wald method), with the results expressed as odds ratios (ORs) and 95% confidence intervals (CIs). Associations with *p* < 0.05 were considered statistically significant.

## 3. Results

A total of 713 women with complete assessments of gestational serum VitD levels and PPD were included in the analyses. Table 1 presents the sociodemographic and gestational characteristics and their relationship with the presence of depression at three months postpartum in the women.

The overall prevalence of postpartum depression (PPD) was 6.9% (n = 49). Regarding serum levels of gestational vitamin D, 31.3% of the women presented deficient values (≤19.9 ng/mL), while 68.7% had non-deficient levels (≥20.0 ng/mL) (Table 1). Most participants were 24 years or older (69.4%), with a higher concentration in the 30 years or older age group (35.6%). More than half of the women belonged to socioeconomic class C (55.0%), followed by classes A/B (27.1%). Regarding pre-gestational nutritional status, women with adequate weight or overweight predominated (66.9%), while 25.2% were obese. Only 3.8% reported a history of psychological or psychiatric treatment. Most participants did not experience an episode of depression during pregnancy (89.6%), and 32.4% reported using vitamin supplementation. Regarding ethnicity, the majority were White (62.7%), and births occurred mainly in winter (37.7%) and autumn (24.4%).

In the bivariate analysis, a higher PPD prevalence was observed among women aged 24 years or older (*p* = 0.037), those from socioeconomic classes D/E (*p* = 0.003), with a history of psychological or psychiatric treatment (*p* = 0.015), and with an episode of depression during pregnancy (*p* < 0.001). Furthermore, deficient VitD levels were also associated with a higher prevalence of PPD (*p* = 0.014). No significant associations were observed between PPD and nutritional status, vitamin supplementation, ethnicity, and season of the year (Table 1).

After adjusting the multivariate model, VitD remained a risk factor for the presence of postpartum depression (PPD). Women with VitD deficiency (levels ≤ 19.9 ng/mL) were approximately twice as likely to develop PPD (Figure 1) compared to those with levels ≥ 20.0 ng/mL (OR = 2.0; 95% CI: 1.0–4.2; *p* = 0.049). Furthermore, other risk factors remained significant in the model, such as maternal age ≥ 24 years, with a risk approximately seven times higher (Figure 1) for women aged 24–29 years (OR = 7.0; 95% CI: 1.5–31.8; *p* = 0.011) and approximately nine times higher (Figure 1) for those aged ≥ 30 years (OR = 9.5; 95% CI: 2.0–43.8; *p* = 0.004); women belonging to socioeconomic classes D/E, who were approximately five times more likely to have PPD (Figure 1) than those in classes A/B (OR = 5.5; 95% CI: 1.8–17.0; *p* = 0.003); and, finally, depression during pregnancy (Figure 1), which also remained an important risk factor (OR = 3.4; 95% CI: 1.5–7.7; *p* = 0.003) (Table 2).

## 4. Discussion

In the present study, vitamin D deficiency during pregnancy was found to be associated with an increased risk of postpartum depression, although with borderline statistical significance (*p* = 0.049) and with less strength compared to the psychosocial predictors included in the model. These results suggest that PPD risk arises from the convergence of biological and psychosocial factors, with vitamin D deficiency acting as a contributing biological factor that may influence neuroimmune and endocrine pathways already activated by social conditions or psychiatric vulnerability [11,25,26].

These findings are consistent with evidence suggesting that vitamin D is a mood modulator during pregnancy and the postpartum period [13,14,27]. Although some of the literature presents inconsistent results on this relationship [28], our findings reinforce the hypothesis that vitamin D deficiency during pregnancy may biologically contribute to the risk of postpartum depression. This deficiency may compromise proper hypothalamic–pituitary–adrenal (HPA) axis modulation, increase NF-κB-mediated inflammatory pathway activation, reduce serotonin and estradiol synthesis, and weaken antioxidant and neuroprotective mechanisms. Together, these physiological changes may destabilize maternal neuroendocrine and affective balance, favoring depressive symptom development in the postpartum period [25].

Although biological mechanisms reinforce the plausibility of this association, the clinical impact tends to vary across populations and contexts [29]. In this sense, the prevalence of PPD in this study was 6.9%, which is lower than the global estimates (17–20%) reported in a meta-analysis by Wang et al. [29], as well as the estimates of 12–25% in Brazil [3]. This discrepancy can be explained by the different methods used to identify PPD, since most epidemiological studies use screening scales, such as the EPDS, PHQ-9, BDI, and CED-S [14], while this study used a structured instrument, allowing diagnostic confirmation according to international criteria and offering a more specific estimate of depression.

Regarding maternal age, women aged 24–29 and ≥30 years presented a significantly higher risk of developing postpartum depression (PPD) compared to the reference group (18–23 years), consistent with large population studies and meta-analyses [30,31]. It is relevant to highlight the remarkably high risk (OR) values observed in our sample for maternal age (24–29 years: OR = 7.0; 95% CI: 1.50–31.8) and ≥30 years (OR = 9.5; 95% CI: 2.0–43.8). Although the risk is real and significant, we should interpret these results with caution, considering the wide confidence intervals, which may have accounted for the few PPD cases in the <23-year group. Nevertheless, the pattern found suggests that advanced maternal age may represent an additional risk marker within a multifactorial vulnerability model for postpartum depression, as per previous studies [30,31].

Socioeconomic status emerged as a strong PPD risk factor, with women from the lowest socioeconomic strata (D + E) showing a five-fold increase compared to the wealthiest strata. This aligns with a recent systematic review by Cooke, et al. [32], which consistently identified low income as also an important risk factor. Financial stressors, limited access to healthcare, and restricted lifestyle options are plausible underlying mechanisms for this association, exacerbating the challenges of childcare in resource-limited settings [9,32,33,34]. Together, these findings reinforce the role of structural socioeconomic inequalities in shaping maternal mental health trajectories during the gestational and postpartum periods.

In addition to socioeconomic vulnerability, a history of depression during pregnancy also played an important role in the development of postpartum depression (PPD) in our sample. These data are consistent with the current literature, which suggests that depression during pregnancy is a significant PPD risk factor [35]. This fact supports the model of depression continuity both during pregnancy and after childbirth, in which PPD reflects the persistence or relapse of a pre-existing depressive episode [36]. In this sense, gestational and postpartum depression should be understood as distinct phases of the same continuous psychopathological process, whose identification during pregnancy is essential for PPD intervention and prevention.

Aligning with this study’s findings, other research has also investigated the relationship between VitD and postpartum depression (PPD). A prospective Australian study revealed that low VitD levels in the second trimester of pregnancy were linked to more than twice the risk of depressive symptoms early postpartum [37]. A 2018 Chinese systematic review corroborated this relationship, indicating that maternal serum VitD concentrations below 20 µg/mL represented a 3.67 times higher risk for developing PPD [13]. However, a study with 2483 women found no significant association between VitD deficiency and PPD, although non-supplemented mothers showed an increase in depressive symptoms after delivery [28]. More recently, a meta-analysis by Centeno et al. [14] confirmed that women with PPD had, on average, VitD concentrations that were about 2.36 ng/mL lower. Despite the consistency of these relationships, the evidence is of low to very low quality due to methodological heterogeneity and differences in diagnostic criteria.

This study has several limitations, including the lack of genetic data related to vitamin D metabolism and dietary intake, as well as the single measurement of serum vitamin D, which may not capture fluctuations due to seasonality or behavioral factors such as sunlight exposure. Skin phototype and daily sunlight exposure were not assessed and may have influenced serum vitamin D levels. Finally, the relatively small number of PPD cases (n = 49) may have reduced the statistical power. Despite these constraints, this study also has notable strengths. Its longitudinal design enabled prospective monitoring from pregnancy to the postpartum period, while the use of a standardized diagnostic instrument ensured greater diagnostic accuracy compared with screening scales. In addition, the sample was representative of urban census tracts in Pelotas, Brazil, and rigorous statistical adjustment for key confounders further strengthened the internal validity of our findings.

Our results reinforce the importance of longitudinally monitoring women’s mental health during pregnancy and the postpartum period. In this sense, recent evidence indicates that depressive symptoms may persist or emerge beyond the first four weeks postpartum, remaining present or fluctuating in intensity until at least six months after delivery [29]. Therefore, we recommend that primary and obstetric care programs consider a surveillance period of at least six months, prioritizing women with multiple risk factors, such as socioeconomic vulnerability, advanced maternal age, history of gestational depression, and deficient serum vitamin D levels [29].

## 5. Conclusions

Our findings suggest an association between maternal vitamin D deficiency during pregnancy and an increased risk of postpartum depression (PPD), although this relationship reached only borderline statistical significance. These results support a multifactorial PPD model, in which social and economic vulnerability; history of depression during pregnancy; demographic characteristics; and biological factors, such as vitamin D deficiency, interact in an integrated manner to modulate the risk of PPD. Within this framework, vitamin D may serve as an additional modulator, particularly among pregnant women exposed to adverse psychosocial conditions. From a clinical perspective, assessing vitamin D status during pregnancy may serve as a simple and low-cost strategy to identify women at higher risk for postpartum depression. Ensuring adequate vitamin D levels through supplementation or lifestyle interventions, in conjunction with identifying and managing psychosocial risk factors, could contribute to improving maternal mental health outcomes and preventing postpartum depression in vulnerable populations. Future studies with larger sample sizes and longitudinal designs are needed to clarify the causal mechanisms involved and to better define the specific role of vitamin D in the etiology and prevention of PPD.

## Figures and Tables

**Figure 1 nutrients-17-03649-f001:**
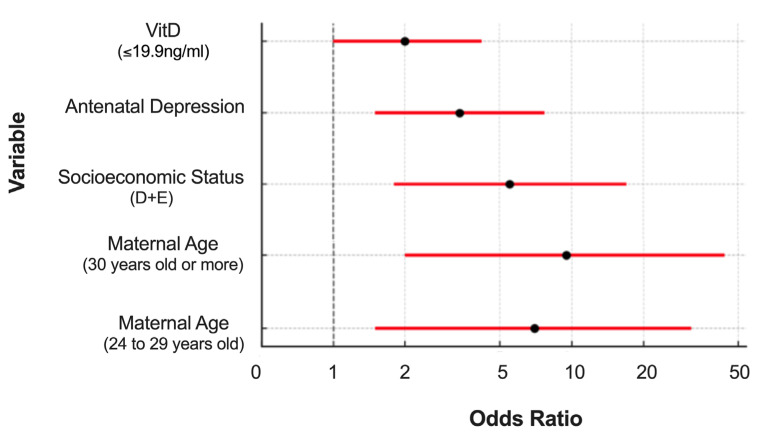
Adjusted odds ratios (ORs) and 95% confidence intervals (95% CIs) for factors independently associated with postpartum depression (PPD) at three months postpartum among women from Pelotas, Brazil. The variables included in the final multivariate logistic regression model are maternal age (24–29 years and ≥30 years), socioeconomic status (D + E), antenatal depression, and gestational vitamin D deficiency (≤19.9 ng/mL). In the plot, the red line represents the 95% confidence interval, the black dot represents the odds ratio, and the dashed vertical line indicates the null value (OR = 1).

**Table 1 nutrients-17-03649-t001:** Sociodemographic, clinical, and gestational characteristics of pregnant women in Pelotas, Brazil, and their association with postpartum depression (n = 713). Data are shown as numbers (n) and percentages (%). Associations were tested via Pearson’s chi-square. *p* < 0.05 was considered statistically significant.

Gestational Variable	Total	PPD (No)	PPD (Yes)	*p*-Value *
	n (%)	n (%)	n (%)	
Age, years				0.037
Up to 23 years old	218 (30.6%)	211 (96.8)	7 (3.2%)	
24 to 29 years old	241 (33.8%)	220 (91.3)	21 (8.7%)	
30 years old or more	254 (35.6%)	233 (91.7)	21 (8.3%)	
Socioeconomic status †				0.003
A + B	193 (27.1%)	185 (95.9)	8 (4.1%)	
C	392 (55.0%)	369 (94.1)	23 (5.9%)	
D + E	116 (16.3%)	100 (86.2)	16 (13.8%)	
Nutritional Status				0.497
Low weight	54 (7.6%)	52 (96.3)	2 (3.7%)	
Healthy weight	258 (36.2%)	237 (91.9)	21 (8.1%)	
Overweight	219 (30.7%)	207 (94.5)	12 (5.5%)	
Obesity	180 (25.2%)	166 (92.9)	14 (7.8%)	
Psychological/Psychiatric Treatment				0.015
Yes	27 (3.8%)	22 (81.5)	5 (18.5%)	
No	686 (96.2%)	642 (93.6)	44 (6.4%)	
Vitamin Supplementation				0.969
Yes	231 (32.4%)	215 (93.1)	16 (6.9%)	
No	482 (67.6%)	449 (93.2)	33 (6.8%)	
Antenatal Depression				<0.001
Yes	74 (10.4%)	58 (78.4)	16 (21.6%)	
No	639 (89.6%)	606 (94.8)	33 (5.2%)	
Ethnic Group †				0.329
White	447 (62.7%)	416 (93.1)	31 (6.9%)	
Black	119 (16.7%)	109 (91.6)	10 (8.4%)	
Other	70 (9.8%)	68 (97.1)	2 (2.9%)	
Season				0.891
Summer	104 (14.6%)	98 (94.2)	6 (5.8%)	
Autumn	174 (24.4%)	163 (93.7)	11 (6.3%)	
Winter	269 (37.7%)	253 (94.1)	16 (5.9%)	
Spring	141 (19.8%)	130 (92.2)	11 (7.8%)	
Gestational VitD				0.014
Deficient (≤19.9 ng/mL)	223 (31.3%)	200 (89.7)	23 (10.3%)	
Not deficient (≥20.0 ng/mL)	490 (68.7%)	464 (94.7)	26 (5.3%)	

Legend: n = absolute number of participants; (%) = proportion in relation to the total category. * Pearson’s chi-square test. † Variables with missing data.

**Table 2 nutrients-17-03649-t002:** Multivariate logistic regression identifying independent predictors of postpartum depression at three months postpartum among women in Pelotas, Brazil. The model was adjusted using the Backward Wald method. Odds ratios (ORs) with 95% confidence intervals (95% CIs) are displayed. Variables excluded from the final model (nutritional status, psychiatric/psychological treatment, vitamin supplementation, ethnicity, and season) did not reach statistical significance and were removed.

Variable	OR	95% CI	*p*-Value
Maternal Age			
Up to 23 years old	1.0		
24 to 29 years old	7.0	1.5–31.8	0.011
30 years old or more	9.5	2.0–43.8	0.004
Socioeconomic Status			
A + B	1.0		
C	1.9	0.7–5.1	0.171
D + E	5.5	1.8–17.0	0.003
Gestational Depression			
No	1.0		
Yes	3.4	1.5–7.7	0.003
Vitamin D			
Not deficient (≥20.0 ng/mL)	1.0		
Deficient (≤19.9 ng/mL)	2.0	1.0–4.2	0.049

Legend: 95% CI = 95% confidence interval; OR = odds ratio. The final adjusted model included the following covariates: maternal age, socioeconomic status, and history of depression during pregnancy.

## Data Availability

The original contributions presented in this study are included in this article. Further inquiries can be directed to the corresponding author.

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
