# Peer review of "Vitamin D Deficiency During Pregnancy Is Associated with Postpartum Depression: A Cohort Study in Southern Brazil"

_nutrients, 2025, doi:10.3390/nu17233649_

Round 1

Reviewer 1 Report

Comments and Suggestions for Authors

Thank you for the opportunity to review this cohort study on Vit D levels and postpartum depression.

The introduction and methods are well written. Rationale of the study is clear.

In the Statistical analysis section (and in table footnotes), please describe the confounders included in the multivariable regression

In the Results sections, Table 1 should present columns that do not have overlapping patients, e.g. with PPD, without PPD, total. The percentage of PPD currently does not make sense in Table 1

The conclusion in the first paragraph of Discussion and the Conclusion section could be moderated by the near-significance relationship (CI includes 1.0, P value approaching 0.05) between Vit D and PPD.

Author Response

Comments 1: Thank you for the opportunity to review this cohort study on Vit D levels and postpartum depression.

Response 1: Thank you very much for your careful review. All your comments and suggestions helped improve the quality of our manuscript.

Comments 2: The introduction and methods are well written. Rationale of the study is clear.

Response 2: We thank the Reviewer for the positive evaluation of these sections. No changes were necessary in response to this comment.

Comments 3: In the Statistical analysis section (and in table footnotes), please describe the confounders included in the multivariable regression.

Response 3: Thank you for your suggestion. We have now clearly described the confounders included in the multivariable logistic regression model. As detailed in the revised Statistical Analysis section (page 4, lines 148-150), the final adjusted model included the following covariates: maternal age, socioeconomic status, and history of depression during pregnancy. These variables were selected based on the theoretical framework and scientific–clinical expertise of the research team. In addition, this information was added to the footnote of Table 2 for clarity.

Comments 4: In the Results sections, Table 1 should present columns that do not have overlapping patients, e.g. with PPD, without PPD, total. The percentage of PPD currently does not make sense in Table 1

Response 4: We appreciate this observation. Table 1 has been reformatted to display three distinct columns—Total, With PPD, and Without PPD—ensuring clarity and preventing overlap. Percentages were recalculated accordingly.

Modification: Results section, page 5; Table 1.

Comments 5: The conclusion in the first paragraph of Discussion and the Conclusion section could be moderated by the near-significance relationship (CI includes 1.0, P value approaching 0.05) between Vit D and PPD.

Response 5: We fully agree. To reflect the borderline statistical significance, we have moderated the language throughout the Discussion and Conclusion sections. For example, the first paragraph of the Discussion (page 7, lines 223–229) now reads as follows:

In the present study, vitamin D deficiency during pregnancy was found to be associated with an increased risk of postpartum depression, although with borderline statistical significance (p = 0.049), and with less strength compared to the psychosocial predictors included in the model.” Similar modifications were applied to the Conclusions (page 9, lines 312–314).

Reviewer 2 Report

Comments and Suggestions for Authors

Authors investigated an association between  Vitamin D deficiency during pregnancy and postpartum depression in Southern Brazil” The study has a clinical relevance and manuscript is well written. There are however some area to improve:

  • Authors use the term “population-based” however the sampling occurred in selected census tracts, not the entire population. Thi8s term is not appropriate and should be replaced, for example “in a cohort of pregnant women in Pelotas.”

  • Introduction seems to long and repeats explaining vitamin D mechanisms; this part should be shortened.

  • Including 244 of 488 women may cause that sample is not representative. This is big limitation to be discussed.

  • Vitamin D was measured only one time but it can change over the time of 24 weeks, please discuss. Vitamin D may depend on sunlight exposure.  Women with very white skin (type 1,2) may be les often under the sun than women with less white skin (type 3,4). This is important limitation and bias.

  • You have OR 2,.00 (UCL is 1.0) and this is a boarder significance p=0.049 implicating low evidence. You conducted multiple testing but did not correct for multiple testing (i.e. Bonferroni correction), After this correction, the OR will become non-significant . Please consult an statistician.

  • How did you handle missing values?

  • Please add further points to limitations: no sunlight exposure or genetics, only a single measurement of vitamin D, small number of PPD cases

Author Response

Comments 1: Authors investigated an association between Vitamin D deficiency during pregnancy and postpartum depression in Southern Brazil” The study has a clinical relevance and manuscript is well written. There are however some area to improve:

Response 1: Thank you very much for your careful review. All your comments and suggestions helped improve the quality of our manuscript.

Comments 2: Authors use the term “population-based” however the sampling occurred in selected census tracts, not the entire population. This term is not appropriate and should be replaced, for example “in a cohort of pregnant women in Pelotas.”

 Response 2: We agree with this observation. The expression “population-based” was replaced with “a cohort study in southern Brazil” throughout the manuscript, including in the title and abstract (Title; Abstract, page 1, line 3; and Section 2.2, page 3, lines 82–84).

Comments 3: Introduction seems to long and repeats explaining vitamin D mechanisms; this part should be shortened.

 Response 3: We have shortened the Introduction by removing redundant explanations of vitamin D mechanisms and summarizing this background more concisely. Approximately 10 lines were removed (page 2, lines 50–57), improving the focus on the study rationale.

Comments 4: Including 244 of 488 women may cause that sample is not representative. This is big limitation to be discussed.

Response 4: We thank the Reviewer for this observation. We would like to clarify that, for the baseline of the cohort, 50% of the 488 census tracts of the urban area of Pelotas were randomly selected in 2016 following a probabilistic design. All households in these selected tracts were visited to identify women up to 24 weeks of pregnancy. Therefore, although not fully population-based, the sampling strategy was random and aimed to ensure representativeness of the city’s urban area.

We also added a clarifying statement in the Methods section:

Section 2.2 (page 3, lines 96–100): “Of the 983 women evaluated at baseline (pregnancy), 755 participated in the post-partum evaluation, totaling 23.2% loss to follow-up. Of these 755, 713 women had complete information on the variables of interest (vitamin D levels during pregnancy and investigation of postpartum depression), corresponding to 72.5% of the initial sample of pregnant women recruited.”

Comments 5: Vitamin D was measured only one time but it can change over the time of 24 weeks, please discuss. Vitamin D may depend on sunlight exposure. Women with very white skin (type 1,2) may be less often under the sun than women with less white skin (type 3,4). This is important limitation and bias.

 Response 5: Thank you for this insightful comment. We have now included this issue in the limitations (Discussion, page 9, lines 292–300): “This study presents some limitations. Additional limitations include the absence of genetic data related to VitD metabolism and dietary intake, the single measurement of serum VitD, which may not capture fluctuations due to seasonality or behavioral factors such as sunlight exposure. Skin phototype and daily exposure to sunlight were not assessed and may have influenced serum VitD levels. Finally, the relatively small number of PPD cases (n=49) may reduce statistical power.”

Comments 6: You have OR 2,.00 (UCL is 1.0) and this is a boarder significance p=0.049 implicating low evidence. You conducted multiple testing but did not correct for multiple testing (i.e. Bonferroni correction), After this correction, the OR will become non-significant . Please consult an statistician.

 Response 6: We thank the reviewer for this insightful comment. We agree that the observed association between gestational vitamin D deficiency and postpartum depression (PPD) presents borderline statistical significance (p = 0.049; 95% CI: 1.0–4.2), indicating weak evidence that must be interpreted with caution. Regarding the Bonferroni correction, we consulted a statistician and confirmed that this adjustment is not applicable in our analytical context, since our model tested a single main hypothesis—the association between vitamin D deficiency and PPD—using a multivariable logistic regression model with pre-specified adjustment variables (maternal age, socioeconomic status, and depression during pregnancy). Bonferroni correction is typically used when multiple independent tests are performed on the same dataset, not for coefficients estimated simultaneously within a single multivariate model that already accounts for confounding. This approach follows established epidemiological recommendations. We greatly appreciate the reviewer’s suggestion; however, after consulting with our statistical team and considering the study design and analytical framework, we respectfully opted to maintain the original analysis, as it remains statistically appropriate and consistent with the study’s main objective.

Comments 7: How did you handle missing values?

 Response 7: The database was processed considering only women who had complete information regarding the primary exposure (vitamin D) and the outcome (postpartum depression); therefore, the main variables of interest do not have missing data. For the other variables with missing data, they were identified and entered into the statistical program, so the results presented already account for missing data.

Comments 8: Please add further points to limitations: no sunlight exposure or genetics, only a single measurement of vitamin D, small number of PPD cases

Response 8: Thank you for this suggestion. We have now included this issue in the limitations (Discussion, page 9, lines 292–300): “This study presents some limitations. Additional limitations include the absence of genetic data related to VitD metabolism and dietary intake, the single measurement of serum VitD, which may not capture fluctuations due to seasonality or behavioral factors such as sunlight exposure. Skin phototype and daily exposure to sunlight were not assessed and may have influenced serum VitD levels. Finally, the relatively small number of PPD cases (n=49) may reduce statistical power.”

Reviewer 3 Report

Comments and Suggestions for Authors

Thank you for the opportunity to review this manuscript. Below are my comments to improve the quality and clarity of the manuscript:

  1. Title: the use of "population-based" may be overstated, consider "community-based cohort" instead.
  2. Add a short line on public health implications in the abstract.
  3. In the introduction, please condense the background of pathophysiology and avoid repetition. Additionally, add a clear final paragraph with a clear study objective.
  4. Please clarify if this is a "nested cohort" or "longitudinal sub-study".
  5. Please report attrition rate, clarify handling of missing data, and discuss possible bias.
  6. Please justify the cut-off value used for vitamin D deficiency, including the reference.
  7. State how confounders were selected and if the model fit was tested.
  8. Important missing factors, such as sunlight and diet, should be stated as limitations.
  9. In the results section, the prevalence of vitamin D deficiency should be written earlier. Use periods for decimals consistently.
  10. Discussion should emphasize "association", not "causation."
  11. The biological explanations should be made more concise and not repetitive.
  12. The discussion section would benefit from adding one short paragraph on policy or clinical implications.
  13. Similarly, add one or two sentences on practical implications in the conclusion. Also, avoid repeating detailed results in the conclusion.
  14. Ensure that there is no duplication in the references, such as #24 and #25.
  15. Also, ensure that all listed authors made substantial intellectual contributions. Contributors involved only in data collection or logistics should be acknowledged rather than listed as authors.

Author Response

Comments 1: Thank you for the opportunity to review this manuscript. Below are my comments to improve the quality and clarity of the manuscript:

Response 1: Thank you very much for your careful review. All your comments and suggestions helped improve the quality of our manuscript.

Comments 2: Title: the use of "population-based" may be overstated, consider "community-based cohort" instead.

Response 2: We agree with this observation. The expression “population-based” was replaced with “a cohort study in southern Brazil” throughout the manuscript, including in the title and abstract (Title; Abstract, page 1, line 3; and Section 2.2, page 3, lines 82–84).

Comments 3: Add a short line on public health implications in the abstract.

Response 3: We added a final sentence to the Abstract (page 1, lines 33–36): “These findings highlight the potential role of VitD in maternal mental health and support the importance of monitoring VitD status during pregnancy. From a public health standpoint, ensuring adequate vitamin D levels in prenatal care may contribute to reducing the burden of postpartum depression.”

Comments 4: In the introduction, please condense the background of pathophysiology and avoid repetition. Additionally, add a clear final paragraph with a clear study objective.

Response 4: The Introduction was condensed to remove repetition and now ends with a clear statement of objectives (page 2, lines 70–73):

“Therefore, this study aimed to investigate the association between maternal vitamin D deficiency during pregnancy and the occurrence of clinically diagnosed postpartum depression three months after delivery, using a longitudinal design with rigorous control for confounding factors.”

Comments 5: Please clarify if this is a "nested cohort" or "longitudinal sub-study".

Response 5: We clarified this in Section 2.2 (page 3, lines 82 – 83): “This is a longitudinal sub-study nested within a larger cohort titled ‘Maternal Neuropsychiatric Disorders in the Pregnancy–Puerperal Cycle.’”

Comments 6: Please report attrition rate, clarify handling of missing data, and discuss possible bias.

Response 6: Thank you for your comment. Of the 983 women evaluated at baseline (pregnancy), 755 participated in the postpartum evaluation, totaling 23.2% loss to follow-up. Of these 755, 713 women had complete information on the variables of interest (vitamin D levels during pregnancy and investigation of postpartum depression), corresponding to 72.5% of the initial sample of pregnant women recruited. This information was added to the manuscript (page 3, lines 96-100). Regarding the missing data, we emphasize that the database was processed considering only women who had complete information regarding the main exposure (vitamin D) and the outcome (postpartum depression); therefore, the main variables of interest do not have missing data. For the other variables with missing data, these were identified and reported in the statistical program, so the results presented already consider the presence of missing data (page 4, lines 153-157).

Comments 7: Please justify the cut-off value used for vitamin D deficiency, including the reference.

Response 7: We thank the reviewer for this valuable comment. We have now added a clear justification and appropriate citation in Section 2.5 (page 4, lines 140–144) to support the cut-off value used for vitamin D deficiency. Serum 25(OH)D concentrations were classified as deficient (<19.9 ng/mL) and not deficient (≥20.0 ng/mL) based on Holick (2007) and further supported by the Endocrine Society Clinical Practice Guidelines (Holick et al., 2011), which represent some of the most widely accepted international references. This threshold corresponds to the serum level required to optimize intestinal calcium absorption and parathyroid hormone suppression, thereby ensuring adequate bone and metabolic health in the general population. Accordingly, the following sentence and reference were added to the Methods section (Biochemical Analysis subsection): “Vitamin D deficiency was defined as serum 25(OH)D levels <20 ng/mL, according to widely accepted clinical practice guidelines, which establish this value as the threshold for maintaining bone and metabolic health.”

Comments 8: State how confounders were selected and if the model fit was tested.

Response 8: A description of the test conducted and its results has been added to the manuscript (page 4, lines 146–152):“The selection of potential confounders for the multivariable logistic regression model was based on the theoretical framework consulted and on scientific and clinical expertise of the research team. The final adjusted model included the following confounders: maternal age, socioeconomic status, and history of depression during pregnancy. Model fit was assessed using the Hosmer–Lemeshow test, which indicated good adequacy across all six stages of the automatic variable selection procedure (p-values ranging from 0.741 to 0.863).”

Comments 9: Important missing factors, such as sunlight and diet, should be stated as limitations.

Response 9: As suggested, these factors were added to the Limitations paragraph (page 9, lines 290–293).

Comments 10: In the results section, the prevalence of vitamin D deficiency should be written earlier. Use periods for decimals consistently.

Response 10: We thank the reviewer for this helpful observation. The prevalence of vitamin D deficiency has been moved to an earlier part of the Results section to improve clarity and flow (page 6, 178 – 188). In addition, all decimal points have been standardized to use periods consistently throughout the text, tables, and figure legends in accordance with the journal’s formatting guidelines.

Comments 11: Discussion should emphasize "association", not "causation."

Response 11: We appreciate the reviewer’s insightful observation. We carefully revised the Discussion section to ensure that all interpretations of our findings emphasize the presence of an association between vitamin D status and postpartum depression, rather than implying a causal relationship. Accordingly, we replaced or rephrased any wording that could suggest causality to accurately reflect the observational nature of our study.

Comments 12: The biological explanations should be made more concise and not repetitive.

Response 12: We thank the reviewer for this valuable suggestion. We have revised the Discussion section to make the biological explanations more concise and to avoid repetition. Redundant sentences were removed, and the key mechanisms linking vitamin D to mood regulation, including its anti-inflammatory, neuroprotective, and serotonergic effects, were summarized more succinctly to improve clarity and flow.

Comments 13: The discussion section would benefit from adding one short paragraph on policy or clinical implications.

Response 13: We thank the reviewer for this valuable suggestion. In response, we have added a paragraph at the end of the Discussion section highlighting the potential clinical and public health implications of our findings (page 9, lines 302 – 309). Specifically, we emphasize that monitoring and managing vitamin D levels during pregnancy could serve as a preventive strategy for postpartum depression, particularly in women with additional psychosocial risk factors. We also note that integrating vitamin D assessment into routine prenatal care may help identify vulnerable women early and promote maternal mental health outcomes.

Comments 14: Similarly, add one or two sentences on practical implications in the conclusion. Also, avoid repeating detailed results in the conclusion.

Response 14: We appreciate the reviewer’s valuable recommendation. Accordingly, we revised the Conclusions section to make it more concise and avoid repeating detailed results. In addition, we incorporated one sentence addressing the practical implications of our findings, emphasizing that regular screening and correction of vitamin D deficiency during pregnancy may represent an accessible preventive measure to support maternal mental health and reduce the risk of postpartum depression (page 9, lines 319 – 325).

Comments 15: Ensure that there is no duplication in the references, such as #24 and #25.

Response 15: We appreciate the reviewer’s careful observation. We have thoroughly reviewed the reference list and removed duplicate entries, including the overlap previously identified between references #24 and #25. The references have been numbered consistently throughout the manuscript.

Comments 16: Also, ensure that all listed authors made substantial intellectual contributions. Contributors involved only in data collection or logistics should be acknowledged rather than listed as authors.

Response 16: We appreciate the reviewer’s thoughtful observation regarding authorship criteria. All individuals listed as authors fulfill the requirements outlined by the International Committee of Medical Journal Editors (ICMJE). Each author made a significant intellectual contribution to the conception and design of the study, data analysis and interpretation, and/or the preparation and critical revision of the manuscript. Those whose involvement was limited to data collection or logistical support were not included as authors.

Round 2

Reviewer 1 Report

Comments and Suggestions for Authors

The authors have addressed my concerns

Reviewer 2 Report

Comments and Suggestions for Authors

-